# Microbial community composition variation in relation to malaria infections in *Junco hyemalis*

**Wilmer Stanley Amaya-Mejia**[1]*, **Ravinder N. M. Sehgal**[2], **Pamela J. Yeh**[1,3]*

**1** Ecology and Evolutionary Biology Department, University of California, Los Angeles, California, United States of America, **2** Biology Department, San Francisco State University, San Francisco, California, United States of America, **3** Santa Fe Institute, Santa Fe, New Mexico, United States of America

* wamayamejia@g.ucla.edu, amayamejiaws@gmail.com (WSA-M); pamelayeh@ucla.edu (PJY)

## Abstract

The association between gut microbiota community composition and parasitic infections can result in complex interactions that impact host health. Determining whether microbiota composition impacts a host's susceptibility to infections or parasitic infections that alter the microbiota will have important implications for mitigation strategies. In this study, we characterized and compared the microbiota of eight wild Oregon Juncos (*Junco hyemalis*) infected with *Plasmodium relictum* GRW04 and eight non-infected individuals. Our results found that alpha diversity was significantly lower in infected birds. Based on beta diversity metrics, compositional turnover was primarily among rare bacterial taxa. The taxa *Muribaculaceae, Acidobacteriales, Pseudomonas* and *Escherichia-Shingella* were depleted in infected birds. When comparing co-occurrence networks of bacterial communities, the microbiota of infected birds had fewer connections and hub taxa were missing compared to non-infected birds. These results show stronger evidence for parasite-mediated microbiota composition and may correspond with characteristics associated with dysbiosis. Through future studies, it may be possible to determine whether these observed changes to microbial communities correlate with alterations to gut microbiota functions or detrimentally impact host health.

## Introduction

The gut microbiota is closely linked with important host functions that can impact ecological interactions. In birds and other animals, these functions can include nutritional uptake, detoxification, immune functions, and potentially serve to outcompete pathogenic microbes [1–7]. Contributions to these functions can vary greatly based on microbial community composition, including bacteria, archaea, and microscopic eukaryotes. In studies of birds, the "core" bacterial composition has been found to consist of *Bacillota* (formerly *Firmicutes*), *Proteobacteria*, *Bacteroidetes*, and *Actinomycetes* (formerly *Actinobacteria)* [8]. The composition of these bacteria is dynamic

**Data availability statement:** All relevant data are within the manuscript. Raw sequencing data are deposited and available online in the SRA (BioProject PRJNA1284973).

**Funding:** This study was supported by the Institute of the Environment and Sustainability, University of California Los Angeles's in the form of a grant awarded to WSA-M (Grant# 403952-SR-19900), the Center for Community Engagement at University of California in the form of a grant awarded to PJY (Grant# 403828-PY-19900), and the University of California, Los Angeles in the form of a salary for WSA-M. The specific roles of these authors are articulated in the 'author contributions' section. The funders had no role in study design, data collection and analysis, decision to publish, or preparation of the manuscript.

**Competing interests:** The authors have declared that no competing interests exist.

[9] with an increase in beneficial bacteria supporting necessary functions. However, significant changes to an established microbiome can also correspond with detrimental effects for host health, known as dysbiosis [10]. Dysbiosis can contribute to direct negative health outcomes in hosts and may indirectly contribute to increased susceptibility to additional pathogens and infections [11,12]. Therefore, understanding how specific life histories or environmental factors alter the gut microbiome composition, we can examine which functions are impacted, assess the impact on host health, and potentially determine whether the gut can facilitate disease mitigation efforts.

"Dysbiosis" is characterized by reduced microbial diversity, reduced abundance of beneficial microbiota, and an increase in the abundance of pathogenic microbiota, known as pathobionts [10,13]. As a result of dysbiosis, studies have shown that hosts experience a reduction in gut microbiome-associated functions. For instance, hosts with lower diversity of strains of *Clostidia*, demonstrated reduced immune functions [14]. Additionally, functions may be dependent on bacteria co-occurrence [13]. In these cases, hosts with reduced bacterial co-occurrence would also experience dysbiosis [15,16]. Therefore, while it can be difficult to pinpoint the cause of dysbiosis in a natural system, it is possible that dysbiosis will correlate with reduced host immunity, increased disease susceptibility and overall disease prevalence [12,17].

A growing body of literature links parasitic infections and host microbiome communities, as reviewed in Palinauskas et al. 2022 [18]. For instance, murine models of malaria infections found that an increase in pathogenic bacteria, *E. coli* O86:B7, primes the immune system and protects against *Plasmodium* infections [19]. In humans, this was linked to the release of anti-α-Gal IgM, which limits the ability of *Plasmodium* to establish in hosts [19,20]. Other studies, again in mice, found that the abundance of *Lactobacillus* and *Bifidobacteria* strongly correlates with resistance to *Plasmodium* infections [21]. Of these, an increase in the abundance of *Lactobacillus* was suggested to decrease the likelihood of *Plasmodium* infections in humans [22]. *Lactobacillus* has additionally been proposed to increase bird's chance of survival following malaria infections [23]. These studies suggest that an increase in beneficial bacteria, and the occasional pathogenic bacteria, can correlate with a *microbiota-mediated regulation of parasitic infections*. However, this relationship may only be apparent in longitudinal studies and may only be relevant when studying the early stages of infection not for overcoming the erythrocytic stages [24].

An alternative may be the inverse relationship, whereby *parasites regulate microbial communities*. Under experimental conditions, malaria infections negatively impacted the microbiome community composition of domestic canaries (*Serinus canaria domestica*) leading to reduced communities and less successful recolonization [25,26]. The results of these experimental studies are further supported by a field study on Eurasian tree sparrows (*Passer montanus*) which also found malaria infection status was negatively correlated with bacterial biodiversity [27]. Under these circumstances, infected birds may be experiencing dysbiosis with associated loss of microbial functions, potentially increasing susceptibility [13]. Continued research is necessary to determine the frequency of these two hypotheses for naturally occurring parasite-microbiota interactions.

Experimental studies are essential for determining causative relationships between parasites and the gut microbiome, but the controlled environments may have limited real-world applications. Thankfully, by studying naturally occurring avian malaria, as has been done for many ecological and evolutionary studies before [28–32], we can compare whether experimental observations are reflected in complex real-world systems. One such system is the population of Oregon Junco (*Junco hyemalis*), also known as Dark-eyed Juncos, in California. Juncos are a species of abundant, small passerine birds located throughout a variety of habitats [33] and are susceptible to *Plasmodium spp.* and closely related parasites [34,35]. Urban populations, in particular, boast large populations across several cities [36] and exhibit low levels of naturally occurring infection prevalence [37]. From a One Health perspective [38], urban habitats are essential for human and wildlife health [39,40], making urban Juncos a biological and medically valuable system.

A few studies have characterized the gut microbiota of wild sparrows. One study specifically compared the microbiota of Juncos, finding no difference associated with sex or habitat. While Juncos in this study retained the core community (*Proteobacteria, Actinomycetes, Bacillota,* and *Bacteroides*), *Proteobacteria* were the most abundant across all individuals [41], similar to results observed in White-crown Sparrows (*Zonotrichia leucophyrs;* Phillips et al., 2018). However, this composition differs from a study on House Sparrows (*Passer domesticus*), which found *Bacillota* to be the most abundant taxa [12]. *Proteobacteria* and *Bacillota* may provide resistance to malaria [18], but whether the composition changes based on infection requires further study. Additionally, the community composition of White-crown Sparrows and House Sparrows were more diverse in urban populations compared to non-urban populations, contrasting the Junco results [12,41–43]. Overall, these results highlight the potential influence of host phylogeny and environmental conditions on gut microbiota as well as the potential significance for disease susceptibility and resistance.

In this study, we characterize the gut microbiota of eight wild Juncos with naturally occurring *Plasmodium relictum* GRW04 infections and eight birds lacking detectable infections. When comparing the microbiota composition based on infection status, we determine whether there is support for 1) the *microbiota-mediated response to infection* hypothesis in which the proportion of beneficial bacteria (*Bacillota Lactobacillus,* or *Bifidobacteria*) will be highest in infected birds or 2) *parasite-mediated gut microbiota* hypothesis in which infected birds demonstrate characteristics of dysbiosis (lower diversity, decreased beneficial bacteria, and increased pathogenic bacteria). As a preliminary study, our results contribute to a growing body of research examining the relevance of microbiota for host health.

## Methods

### Sample collection

The Junco breeding populations in urban habitats across southern California were captured via mist netting with audio lures in 2022 and 2023. Individual birds were targeted for a maximum of 30-min intervals. Birds were banded with unique leg band combinations consisting of one metal federal aluminum band and three plastic color bands. Body morphometrics and characteristics, including weight, wing cord length, tail length, tarsus length, bill measurements, sex, and age, were recorded [44]. Body condition was calculated based on the residuals of a linear regression between tarsus length and body weight [45]. Blood samples, < 1% of the bird's body mass, were collected via brachial venipuncture with a 30G needle, transferred using heparinized capillary tubes to prepare blood smears and the remaining blood (approximately 50 μL) was stored in lysis buffer until processing [46]. Cloacal swabs were used as a non-invasive approach to sample the gut microbiome [47]. The area around the cloaca was initially sterilized with an alcohol pad to ensure no feathers were in contact with the surface. The area was allowed to dry before inserting a FLOQSwab 516CS01 Ultra Minitip Flocked Swab (Copan, Murrieta, CA, USA). Cloacal swabs were twisted clockwise two times before removal, and samples were transferred directly into RNAlater (Fisher Scientific, Hanover Park, IL, USA) for storage at 4 °C in the field. RNAlater-buffered samples were later stored at −80 °C before extraction. All birds were released following processing.

## Ethics approval statement

All animal handling in this study was approved by the Institutional Animal Care and Use Committee (IACUC) of UCLA (ARC-2018–007-AM-004), followed relevant ARRIVE methods [48], and did not involve anesthesia, analgesia, or endangered/protected species. Banding efforts were conducted in compliance with the Ethics and Responsibilities of Bird Banders published by the US Geological Survey Federal Bird Banding Laboratory (Permit #23809) and as outlined by the State of California Department of Fish and Wildlife Scientific Collecting Permit – Specific Use (S-191300002-20288-001-02) for taking/possession of wildlife for scientific purposes.

## Blood sample preparation

We prepared two blood smears from each individual. Smears were air dried in the field and fixed in methanol the same day. The fixed blood smears were stained within 30 days of collection with Shandon Wright-Giemsa stain kits following manufacturer protocols (Thermo Fisher Scientific, Kalamazoo, MI, USA). Blood smears were examined using an Olympus CH2 light microscope. Each blood smear slide was manually inspected under x1000 magnification to quantify the number of infected and non-infected red blood cells (RBC) out of 10,000 RBCs [49]. A representative slide shows the macrophage of *P. relictum* GRW04 in the peripheral blood of a Junco, band number 2881–56441 collected in Los Angeles on April 20th, 2023 (Fig 1). The correlation between body condition and parasitemia was calculated for all birds based on a Pearson's product-moment correlation in R 4.4.2 [50].

Blood samples were used for DNA extractions with the DNeasy Blood and Tissue kit following the manufacturer's protocol (Qiagen, CA). DNA extractions were assessed with a Nanodrop ND-1000 UV-Vis Spetrophotometer (ThermoFisher, MA, USA) prior to screening for the presence of *Plasmodium relictum GRW04* infections using a nested PCR approach [51]. This nested PCR approach consists of an initial PCR with the primers HaemNF (CATATATTAAGAGAATTATGGAG) and HaemNR2 (AGAGGTGTAGCATATCTATCTAC) performed with 20x cycles followed by the primers HaemF (ATGGT-GCTTTCGATATATGCATG) and HaemR (GCATTATCTGGATGTGATAATGGT) performed with 35x cycles. This technique amplified a fragment of the mitochondrial *cty b* gene. The PCR mix included 12.5 µL of 2X DreamTaq PCR MasterMix (Thermo Fisher Scientific Baltics, Vilnius, LT) 6.5 µL nuclease-free $H_2O$, 0.5 µL of each primer, and 5 µL of template DNA

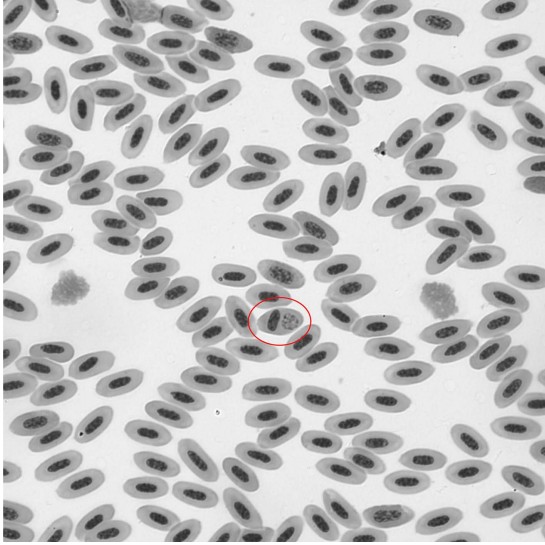

**Fig 1. *Plasmodium relictum* GRW04-infected erythrocyte of Junco.**

(~100ng/uL). The nested PCR product was visualized on a 2% agarose gel with 5 μL of the nested PCR product. Positive samples were cleaned using ExoSapIT following manufacturer's instructions and submitted to GENEWIZ Sanger sequencing on the Applied Biosystems platform. Only sequences with >80% high-quality reads were used for analysis. Using BLAST, the high-quality sequence was compared against the GenBank and MalAvi database. Lineages where able to be identified as *Plasmodium relictum* GRW04 when lineages were identical or had ≤ 1 bp differences [52]. After screening blood samples, only cloacal swabs obtained from birds with clear presence/absence of the lineage *Plasmodium relictum* GRW04 were subsequently sequenced. Additional details on the haemosporidian prevalence and diversity can be found [37].

## Cloacal swab preparation and processing

When possible, cloacal swabs were collected for DNA extraction. We isolated cloacal genomic DNA from 16 males found in urban habitats (across Los Angeles and San Diego), the youngest of which were in their second year. This included eight *Plasmodium relictum GRW04*-infected and eight non-infected birds. To account for between-season variation, samples were only compared when collected from birds during the one breeding season [53]. Our primary hypothesis concerned the abundance of *Bacillota*, which can vary due to environmental factors between 10–20% [12,43]. Therefore, we performed a power analysis based on a difference of >10%, finding our sample size would be sufficient for an initial comparison. DNA extractions were performed on cloacal swabs using the Quick-DNA Fungal/Bacterial Miniprep Kit following a modified protocol (ZYMO Research, Irvine, CA, USA). The main modification to the standard protocol was elution into molecular grade nuclease-free water rather than the provided elution buffer. The elutions were subsequently evaporated at 37 °C in10 minute intervals until the final volume was ∼50 μL to increase sample concentration. Genomic DNA quality was assessed using NanoDrop One (OD260/280 between 1.8–2.0) to verify >200ng of DNA at a concentration of 20ng/ μL. Samples were submitted to Novogene for 16S Amplicon Sequencing of the V3-V4 region, primers 341F [5'-CCTAYGGGRBGCASCAG-3'] and 806R [5'-GGACTACNNGGGTATCTAAT-3'], sequenced on an Illumina NovaSeq 6000 with 100K raw tags (Sacramento, CA, USA).

Sequencing reads were analyzed with the QIIME qii2-amplicon-2024.2 plug-in [54]. Paired-end reads were processes using the DADA2 pipeline [55] (via q2-dada2) and taxonomy assigned for the resulting amplicon sequence variants (ASVs) (via q2-phylogeny) using the Naïve Bayes classifier [56] (via q2-feature-classifer) that was trained on the SILVA 138 database [57].

We compared core bacterial communities, calculated alpha diversity, beta diversity (via q2-diversity), and differential abundance of bacterial taxa, based on rarefied ASVs, for samples based on infection status (via q2-composition) [58]. Core communities were calculated based on relative abundance at the phylum level. Differences in core communities was calculated using the Mann-Whitney U test with Benjamini-Hochberg adjusted *p*-values (*q*-value, α = 0.05). Alpha diversity analysis included Shannon's diversity index, Faith's phylogenetic diversity, and Pielou's Evenness [59] and was compared with a pairwise Kruskal-Wallis test. For beta diversity, Jaccard and Bray-Curtis dissimilarity indices were calculated after samples were rarefied to measure qualitative and quantitative community dissimilarity respectively and compared using a PERMANOVA test with 999 permutations. To determine whether the taxa *Bacillota* varied between groups, we performed a Wilcox rank sum test based on relative abundance. We also compared the effect of parasitemia on each diversity metric to examine potential dose-effects. For this, we checked for normality using the qqp function in "car". We subsequently ran separate linear models for each diversity metric with parasitemia as the effector variable. Finally, differential abundance was tested using the analysis of compositions of microbiomes with bias correction (ANCOM-BC) at the Family and Genus level. Taxa were considered to be significantly depleted or enriched when log-fold change (LFC)> |2| and had a false discovery rate corrected *p*-value < 0.05 (*q*-value).

Co-occurrence microbial networks were assembled for each treatment group to assess bacterial community composition. Bacterial taxa were used to define network nodes, and edges were determined based on significant positive or

negative co-occurrence interactions between nodes (weight > |0.8|). Value calculations and network construction were performed with Sparse Correlations for Compositional data (SparCC) in R [15]. The Gephi 0.10.1 program [60] was used to visualize the networks and to measure topological features, including: number of nodes and edges, network diameter, average degree, weighted degree, average path length, modularity, number of modules, and eigenvectors.

Lastly, network characteristics were compared via the NetCoMi package in R [16]. Networks were initially constructed based on the top 100 most frequent ASVs in each treatment group, measured with Pearson's correlation test, normalized with Central Log Ratio, zero handling with multiplicative imputation, and a threshold set to 0.3. Modularity clusters were optimized using the fast greedy algorithm and the final networks were compared with 1000 permutations; outputs considered global network properties and the most central nodes in the network.

## Results

Our results represent an exploratory study into the relationship between malaria and gut microbiota of free-living Juncos. The peripheral blood of wild Juncos was screened for *P. relictum* GRW04 infections across southern California. Infection prevalence in urban centers was 17% (25/151) and we processed eight individuals without detectable infections and eight with infections [37]. Based on microscopy, parasitemia for these eight infected birds ranged from 0.1–0.7% infected RBC. While *P. relictum* parasitemia can exceed 50% in severe cases, wild-caught birds often have low levels [61], and across studies, we have typically observed ~1% in wild juncos [35,37]. Body condition, based on residuals of tarsus length and body mass, increased significantly with parasitemia (Pearson's product-moment correlation: 0.099, $p = 0.018$). However, based on our study with a larger sample size across multiple years (n = 542), we did not find a correlation between infection status and body condition [37], suggesting this may be a result of our current sample size. A full breakdown of individual metadata, including parasitemia, is provided in Table 1.

Using cloacal swabs as a proxy for gut microbiota composition, we analyzed the infected and non-infected juncos' ASVs. While relative abundance of ASVs appeared to differ between individual samples (Fig 2a), these differences were not statistically significant based on infection status (Fig 2b). The core bacteria for non-infected and infected birds consisted of the four expected taxa: *Actinomycetes* (non-infected: μ = 30.5, σ = 34.5; infected: μ = 42.5, σ = 30.4, Wilcoxon adj.p-value = 0.321), *Bacteroidota* (non-infected: μ = 10.6, σ = 9.43; infected: μ = 5.25, σ = 9.21, Wilcoxon adj.p-value = 0.161), *Bacillota* (non-infected: μ = 20.7, σ = 28.2; infected: μ = 10.6, σ = 15.9, Wilcoxon adj.p-value = 0.382), *and Proteobacteria* (non-infected: μ = 23.3, σ = 17.7; infected: μ = 33.1, σ = 31.6, Wilcoxon adj.p-value = 0.536). *Cyanobacteria* were abundant in a few samples (Fig 2) but were not considered core bacterial taxa.

The correlation between infection status and bacterial diversity were calculated for alpha and beta diversity metrics. For alpha diversity (Fig 3), observed features (Kruskal-Wallis test, $q = 0.92$) and evenness (Kruskal-Wallis test, $q = 0.25$) did not significantly differ between samples based on infected status. However, phylogenetic dissimilarity, Faith's PD, was significantly higher (Kruskal-Wallis test, $q = 0.04$) in control birds compared to *P. relictum* GRW04 infected birds. For beta community composition, the diversity of ASVs, Jaccard Dissimilarity, was significantly different between infected and control birds (PERMANOVA, $q = 0.004$; F = 1.75) while the relative abundances (Bray-Curtis Dissimilarity) did not differ based between infection status (PERMANOVA, $q = 0.58$, F = 0.84). To determine whether parasitemia may have a dose effect, we ran a linear model for each diversity metric but did not find any correlation between parasitemia and diversity (observed features, β = −39.12 ± 62.18, $t = −0.619$ $p = 0.547$), (phylogenetic distance, β = −9.15 ± 7.64, $t = −1.196$, $p = 0.255$), (evenness, β = −0.11 ± 1.18, $t = −0.58$ $p = 0.567$), (Jaccard distance, β = −0.38 ± 0.23, $t = −1.658$, $p = 0.123$), (Bray-Curtis Dissimilarity, β = 0.11 ± 0.26, $t = 0.426$, $p = 0.678$).

The differential abundance, based on ANCOM-BC, consistently showed depletion of ASV in *P. relictum* GRW04-infected birds (Fig 4). When comparing abundance at the family level, six ASVs were depleted in infected birds relative to non-infected birds (*Muribaculaceae*, LFC = −3.85, $q = 0.0001$; *Acidobacteriales*, LFC = −3.14, $q < 0.0001$; *Pseudomonadaceae*, LFC = −3.09, $q < 0.0001$; *Lachnospiraceae*, LFC = −3.90, $q = 0.0002$; *Tannerellaceae*, LFC = −2.95, $q = 0.0003$;

**Table 1. Sample metadata. Band number corresponds with USGS federal aluminum band number.**

| Sample # | Capture Date | USGS Band# | Infection Status | Parasitemia | Tarsus (mm) | Wing (mm) | Body Condition | GPS Coordinates | Extraction Date |
|---|---|---|---|---|---|---|---|---|---|
| N1 | 4/15/2022 | 1821-27611 | Not detectable | 0 | 17.5 | 74 | −0.10365 | 34.071679, −118.440327 | 4/10/2023 |
| N2 | 4/15/2022 | 1821-27490 | Not detectable | 0 | 18.6 | 73 | −0.19852 | 34.071963, −118.439872 | 4/10/2023 |
| N3 | 4/14/2023 | 2881-56431 | Not detectable | 0 | 17.7 | 76 | 0.342736 | 34.0731871, −118.4493052 | 4/27/2023 |
| N4 | 3/17/2023 | 1821-27399 | Not detectable | 0 | 18.5 | 72 | −0.57171 | 34.0675490, −118.4444085 | 5/18/2023 |
| N5 | 4/18/2023 | 2881-56435 | Not detectable | 0 | 19.6 | 72 | −0.56657 | 34.0719474, −118.4380547 | 6/1/2023 |
| N6 | 5/22/2023 | 880-70566 | Not detectable | 0 | 19.8 | 77 | −0.62018 | 34.0712647, −118.4445175 | 5/23/2023 |
| N7 | 5/26/2022 | 1821-27640 | Not detectable | 0 | 19 | 76 | −0.30574 | 34.0729877, −118.4531653 | 11/8/2023 |
| N8 | 2/2/2023 | 880-70579 | Not detectable | 0 | 17.7 | 77 | 0.342736 | 34.0666805, −118.4402246 | 11/8/2023 |
| P1 | 3/27/2023 | 2881-56404 | Detectable | 0.5 | 18.4 | 75 | 1.855096 | 32.8798656, −117.2374852 | 11/8/2023 |
| P2 | 4/20/2023 | 2881-56441 | Detectable | 0.5 | 19.8 | 74 | 1.179816 | 34.0686200, −118.4471520 | 5/24/2023 |
| P3 | 4/20/2023 | 1821-27346 | Detectable | 0.2 | 19.8 | 72 | −0.22018 | 34.070033, −118.451156 | 5/24/2023 |
| P4 | 6/26/2023 | 1821-27207 | Detectable | 0.7 | 20 | 73 | 0.426204 | 34.0710356, −118.4434748 | 11/8/2023 |
| P5 | 6/1/2023 | 2881-56310 | Detectable | 0.3 | 18 | 76 | −0.83768 | 34.0717210, −118.4537449 | 6/7/2023 |
| P6 | 6/8/2023 | 1821-27379 | Detectable | 0.1 | 17.7 | 69 | −0.25726 | 34.0735657, −118.4517705 | 11/8/2023 |
| P7 | 6/16/2023 | 1821-27358 | Detectable | 0.2 | 19.1 | 75 | −0.53254 | 34.0356776, −118.4789245 | 11/8/2023 |
| P8 | 6/29/2023 | 2881-56338 | Detectable | 0.1 | 19.1 | 72 | 0.067456 | 34.0331917, −118.3137598 | 11/8/2023 |

Parasitemia is based on # of infected red bloods/1000. Body condition is the residuals of tarsus length (mm) and body mass (g).

*Saccharimonadaceae,* LFC = −2.29, $q$ = 0.0003). At the genus level, only four bacterial taxa were found to differ based on infection status, all of which were depleted (*Muribaculaceae,* LFC = −3.88, $q$ = 0.001*; Acidobacteriales,* LFC = −3.1.7, $q$ = 0.001*; Escherichia-Shigella,* LFC = −3.97, $q$ = 0.014*; Pseudomonas,* LFC = −3.13, $q$ = 0.02)

Bacterial co-occurrence networks differed between infected and non-infected birds (Table 2). The networks of infected birds had fewer nodes and edges than control birds (Fig 5). Infected birds also had higher modularity, dividing communities into more clusters. Control birds had a central cluster of bacteria with high eigenvector values, including three differentially abundant bacteria (Fig 6). Many significant clusters were inconsistent between samples (Fig 5). While global network properties were not statistically different between infected and non-infected birds, specific node characteristics differed significantly. We found the degree (number of connections between nodes, Jaccard index = 0.16, $p$ = 0.046), betweenness centrality (nodes on the shortest path, Jaccard index = 0.15, $p$ = 0.028), and hub taxa (central nodes, Jaccard index = 0.000, $p$ = 0.039) were more dissimilar than expected by chance. However, neither closeness centrality (proximity between nodes, Jaccard index = 0.19, $p$ = 0.072) nor eigenvector centrality (relative significance of a node, Jaccard index = 0.23, $p$ = 0.185) differed significantly between birds based on infection status. These results were not associated with any specific ASV.

## Discussion

The results of this study add to a growing body of work linking malaria with changes to the host microbial composition [12,18,19,21,24–27]. These previous studies include experimental and field observations with relatively consistent results despite differences in host-parasite combinations. For our contribution, we specifically focused on the Junco, a small passerine native to the Americas [33], which has recently become established in urban centers throughout California [36,62]. In addition to their high abundance and proximity to humans, Juncos and other bird species can contribute to local avian and zoonotic bacterial pathogen systems [35,63]. By characterizing the gut microbiota of Juncos, this work, and potentially future studies in other urban bird species, can provide insight into the relationship between parasites and host bacterial communities.

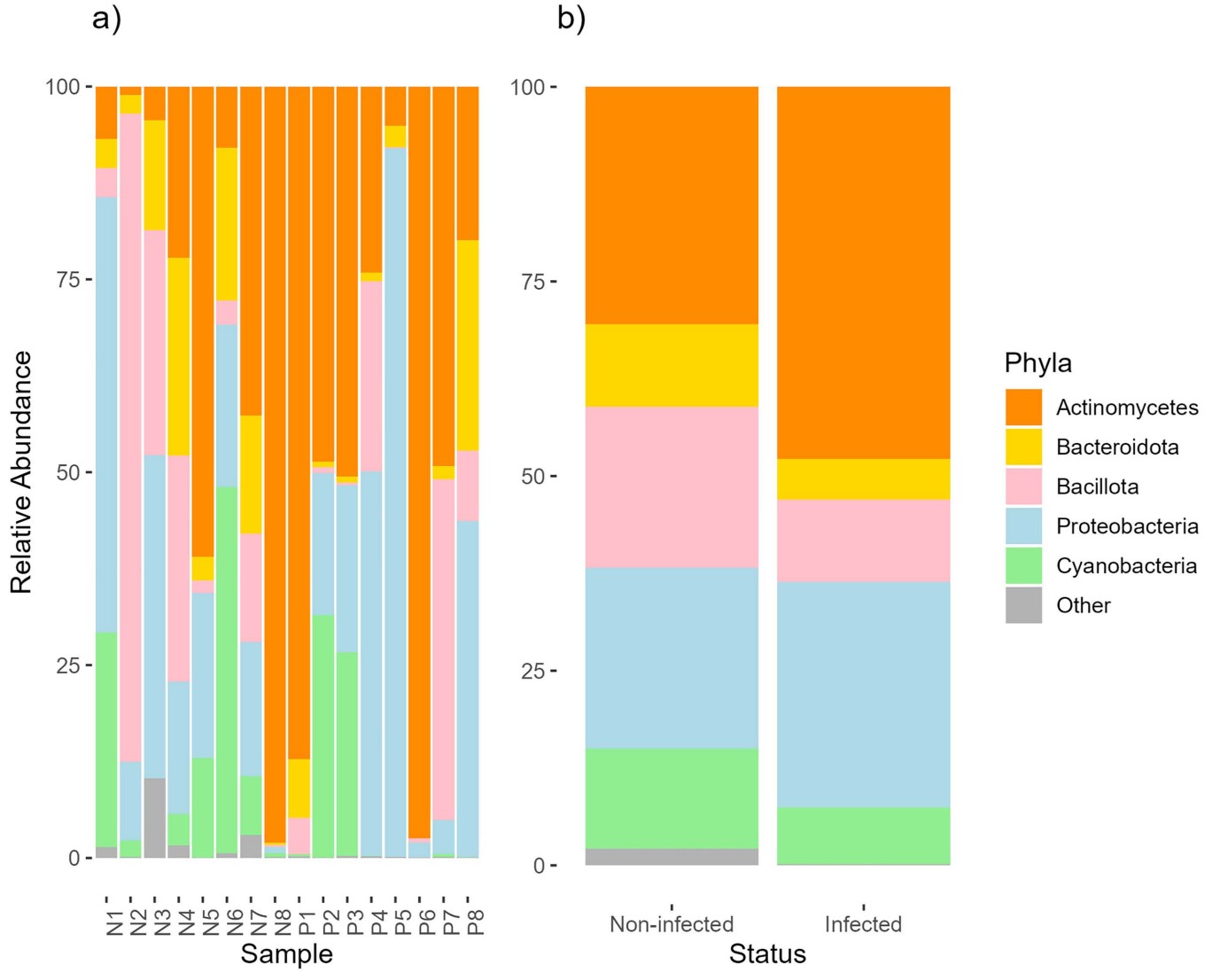

**Fig 2. Core bacterial community composition.** Plots show **a)** per individual sample variation and **b)** pooled samples based on *Plasmodium relictum* infection status. "Core" bacterial are based on presence across all samples and includes: *Actinomycetes* (orange), *Bacteroidota* (yellow), *Bacillota* (pink), and *Proteobacteria* (light blue). *Cyanobacteria* (light green) were prominent in a few samples and were abundant but were not considered as "core" bacteria.

We aimed to determine whether parasitic infections correlated with significant changes to the gut microbiota community of Juncos. If the relative abundance of beneficial bacteria (i.e., *Bacillota, Lactobacillus, Bifidobacteria* [18]) was higher in infected birds, our results would support the hypothesis that *gut microbiota can mediate host response to infection.* In contrast, if infected Juncos had characteristics of dysbiosis (lower biodiversity, lower abundance of beneficial bacteria, higher abundance of detrimental bacteria [13]), our results would support the hypothesis of *parasite-mediated changes to microbiota community composition.* Based on our results, and considering the conclusions from prior studies, we report there is potential support for *parasite-mediated changes to microbiota community composition*.

Based on multiple metrics of biodiversity, the gut microbiota of malaria-infected Juncos was slightly less diverse compared to non-infected Juncos. Unsurprisingly, at the broadest level, our Mann-Whitney U test did not find any significant

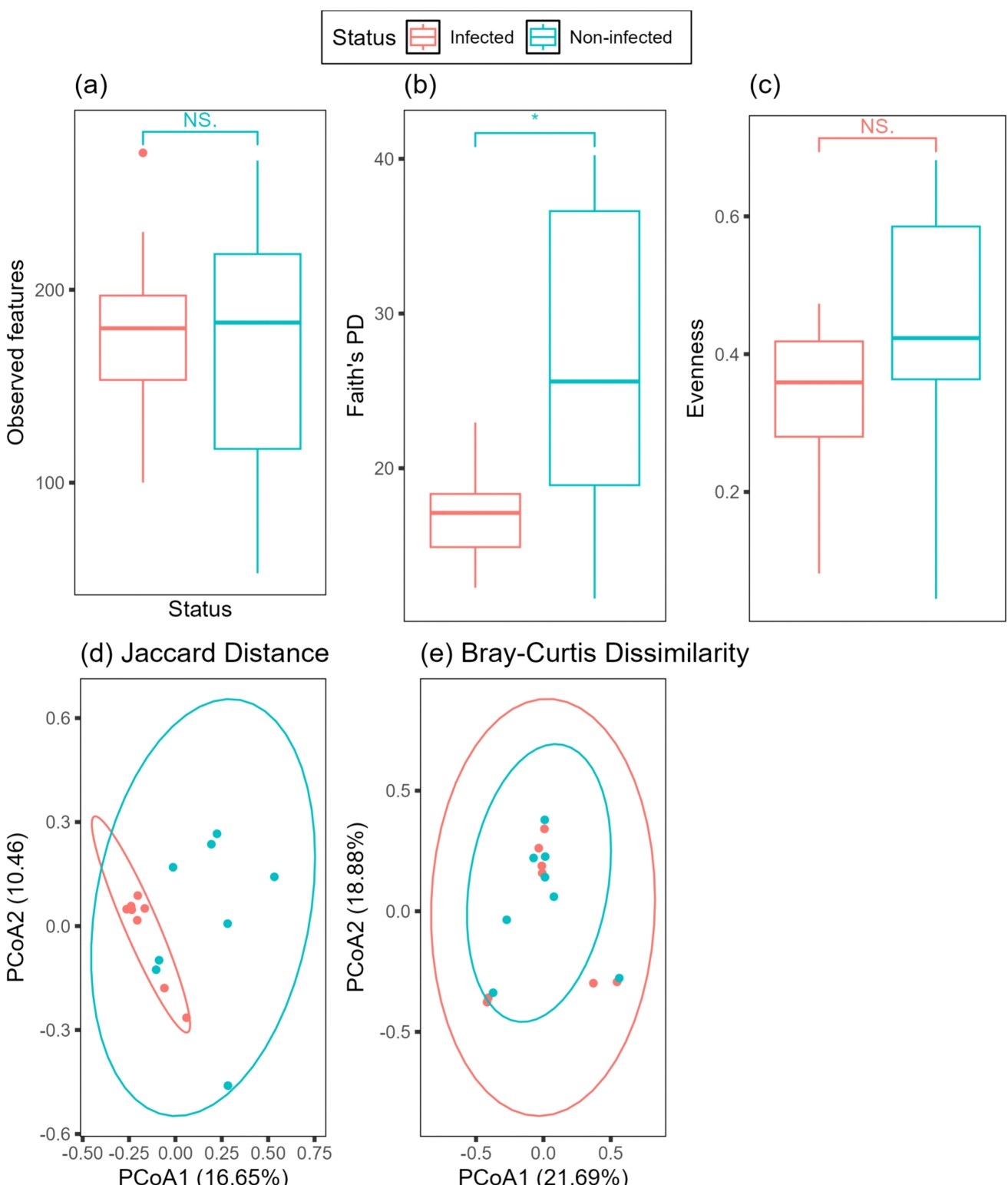

**Fig 3. Bacterial community diversity indices of *Junco hyemalis*.** Individuals infected with *Plasmodium relictum* GRW04 are shown in red, individuals not infected are shown in blue. Alpha diversity includes **a)** observed features, **b)** Faith's phylogenetic diversity, and **c)** Pielou's evenness. Beta diversity includes **d)** Jaccard Distance and **e)** Bray-Curtis Dissimilarity indices. Faith's PD (*q=0.04*) and Jaccard Distance (*q=0.004*) were significantly different between samples.

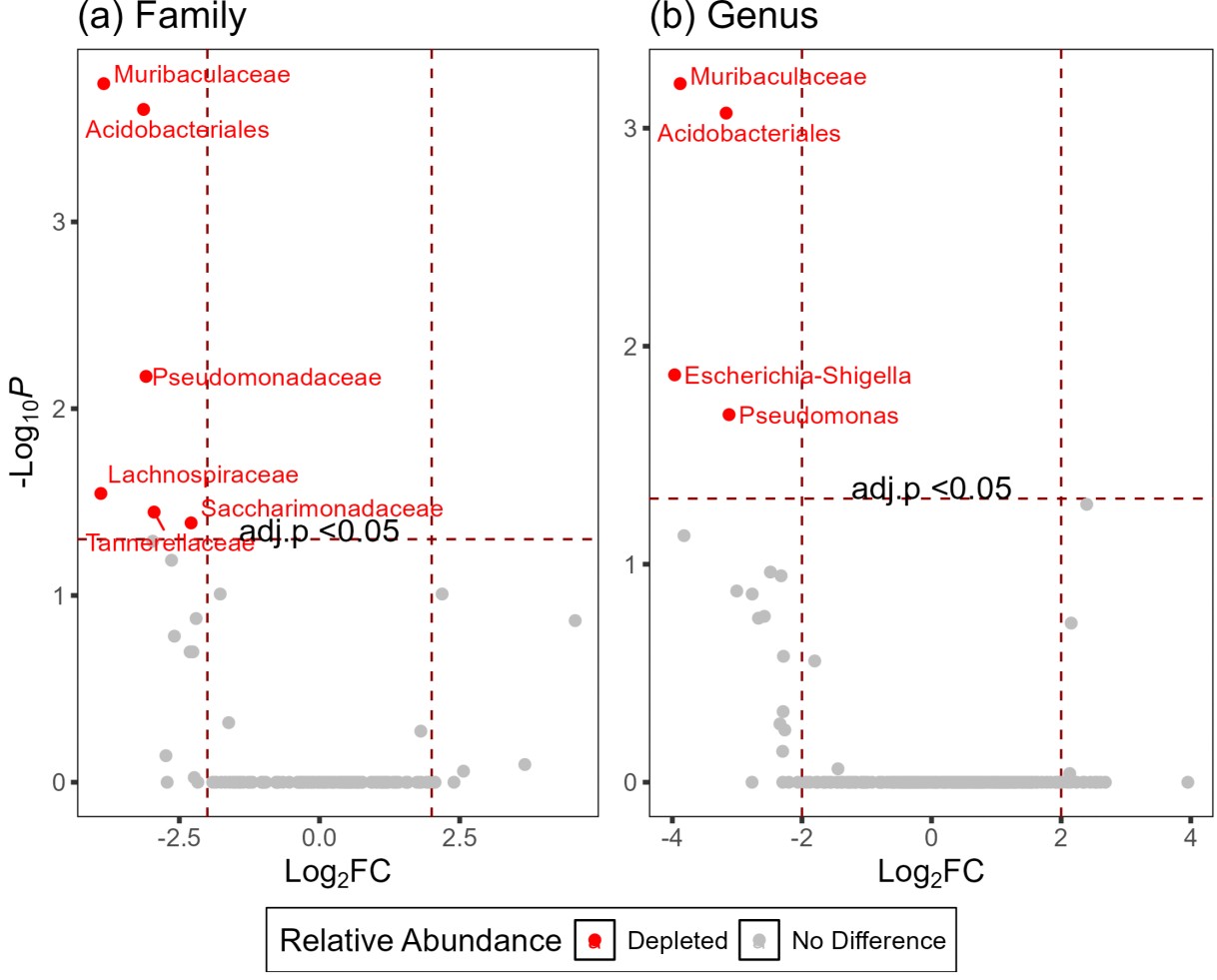

**Fig 4. Comparison of differential abundance with ANCOM-BC.** Differentially abundant taxa at **a)** Family and **b)** Genus level in *Plasmodium relictum* GRW04-infected birds relative to non-infected birds.

difference in the core microbial community composition based on infection status. While it is possible for infections to correlate with broad changes in the community composition of gut microbiota, this is primarily reported in mammal models and not observed in avian systems [25–27,64,65]. When comparing alpha and beta community compositions, our results differed based on the statistical analyses, a common limitations attributed to diversity metrics [66]. For alpha diversity, Faith's PD was lower in infected birds and for beta diversity, Jaccard Dissimilarity values differ between groups. As biodiversity metrics that accounted for relative abundance (evenness, observed features, Bray-Curtis Dissimilarity) did not differ, our results suggest that malaria infection may impact microbiota by reducing the presence of rare bacteria, though additional studies are necessary to support this claim.

Previous studies found the relative abundance of specific bacterial taxa could vary based on infection status [25–27], and our results were partially consistent with these observations. Across all four studies, two potentially pathogenic bacteria (*Pseudomonas* and *Escherichia-Shingella*) were consistently found to be depleted in infected birds. Pathogenic bacteria have been reported to provide immune-related functions by priming the immune system [19,20]. Therefore, while the depletion of these bacterial taxa may not be a traditional metric, the potential loss of function could

**Table 2. Topological parameters of co-occurrence networks.**

| Network Features | Non-infected | *P. relictum* GRW04-infected |
|---|---|---|
| Nodes | 129 | 98 |
| Edges | 279 | 126 |
| Positive | 228 (81.7%) | 71 (56.3%) |
| Negative | 51 (18.3%) | 55 (43.7%) |
| Network diameter | 10 | 9 |
| Average degree | 4.33 | 2.57 |
| Weighted degree | 2.38 | 0.29 |
| Average path length | 4.50 | 3.93 |
| Modularity | 0.78 | 3.48 |
| Number of modules | 37 | 44 |
| Average clustering coefficient | 0.55 | 0.43 |

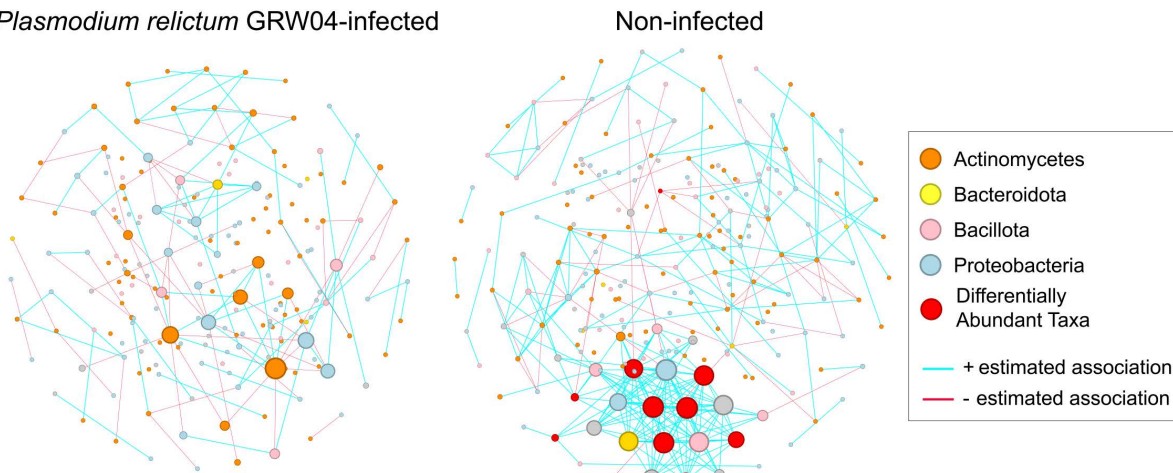

**Fig 5. Bacterial co-occurrence networks.** Samples were grouped based on presence (left) or absence (right) of *Plasmodium relictum* GRW04 infections. Individual nodes represent an amplicon sequence variants and size is based on Eigenvector values. Edges represent positive (blue) or negative (red) interactions between one or more nodes. ASV designated as Firmicutes are shown in yellow. Differentially abundant taxa were depleted in infected samples.

qualify this change as a characteristic of dysbiosis [13]. When examining the relative abundance of potentially beneficial taxa, *Bacillota* and *Lactobacillus,* our study differed from previous reports [21,23]. We did not observe any difference in either taxon (Fig 4), which may may be due to sample size, but we did find that *Muribaculaceae* was depleted in infected birds. *Muribaculaceae* and *Bacillota* are reportedly redundant, sharing beneficial functions within a host, and thus, the difference observed in our study may be reflective of host-specific variation [67], a potential area of future study. Lastly, *Acidobacteriales* was depleted in infected birds, but these could not be classified beyond the order, limiting our ability to interpret the significance of this difference. In their prior study, Van Veelen *et al.* (2017) reported *Acidobacteriales* as a core taxon of woodlarks (*Lullula arborea*) and skylarks (*Alaudia arvensis*) and suggested these are related to nesting substrates [68]. This may represent a potential implication linking malaria and parental investment, if verified. More generally, our results, corroborated by similar studies, highlight a potentially antagonistic interaction between malaria a subset of gut microbes.

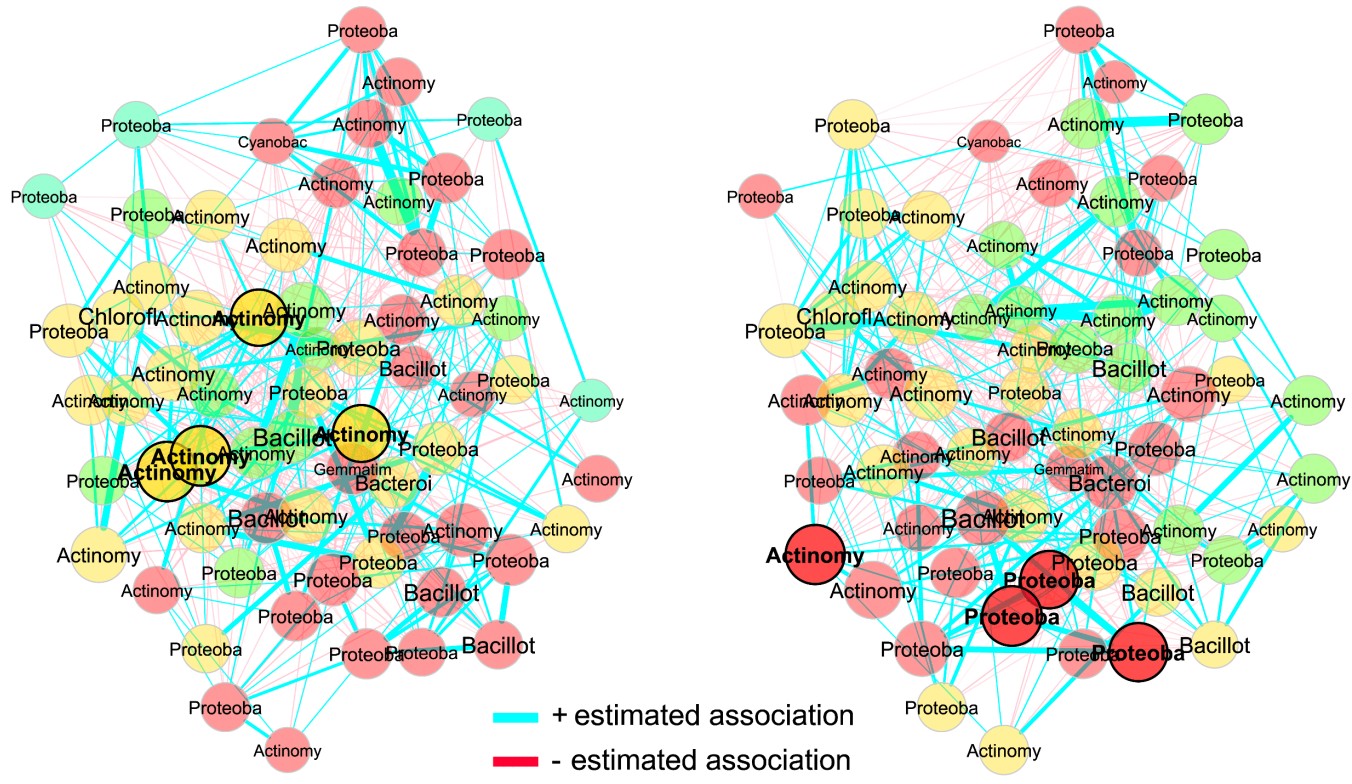

**Fig 6. Normalized bacterial co-occurrence networks.** Samples were grouped based on presence (left) or absence (right) of *Plasmodium relictum* GRW04 infections. Nodes were normalized based on position in non-infected network. Edges represent positive (blue) or negative (red) interactions between one or more nodes. Nodes colors represent clusters.

We examined whether there was a dose effect of infection, parasitemia, on microbiota community composition and relative abundance of bacterial taxa, and host body condition. We did not observe a correlation between parasitemia and alpha or beta diversity metrics. This was likely a sample size issue given the high variability between samples. When comparing parasitemia and bacterial taxa, *Escherichia-Shingella* and *Muribaculaceae* were entirely depleted in infected birds, preventing statistical comparison and relative abundance of *Pseudomonas* did not correlate with parasitemia. Lastly, we did observe a positive correlation between body condition and parasitemia. As a prior study on these populations did not report any correlation between infection status and body conditions [35], it is more likely this correlation was again due to our sample size. However, the previous study did not account for parasitemia, and as *Plasmodium* infections can result in organ swelling [49], it may be possible that infected birds are experiencing an increase in mass relative to their body size. While our results suggest *Plasmodium* infections are correlated with metrics of dysbiosis, based on our dataset, we cannot determine whether infections, changes to microbiota communities, or loss of bacterial functions correspond with a systemic impact on a host.

The co-occurrence networks provide our final support for the *parasite-mediated change to microbiota* hypothesis. The co-occurrence networks were significantly dissimilar based on infection status, more than is expected by random chance. Specifically, the bacterial communities in infected birds appear to have fewer interactions and important taxa are missing (Figs 5, 6). Co-occurrence may be necessarily for bacterial functions [69], and may reflect another metric for dysbiosis. Considering *Plasmodium* infections have also been negatively correlated with bacterial co-occurrence in other studies [25] with prolonged detrimental impacts [26], it would be valuable to determine which bacterial community functions, if any, are impacted.

The gut microbiome contributes significant functions to the host. Based on our results, *Plasmodium relictum* GRW04 was negatively correlated with some metrics of gut microbiota communities in Juncos. As we only considered the impact of infection on certain metrics of biodiversity, it would be important to determine whether these changes are indicative of changes to bacteria functions, as this is the principle outcome of dysbiosis [13]. Even if infections correlate with dysbiosis, it would also be necessary to determine whether this state results in detrimental health outcomes. Through experimental and field studies, we are beginning to understand the complex relationships between the gut microbiome and parasites with exciting new research questions that could potentially guide future mitigation strategies.

## Limitations

As with many field studies on microbiota, there are important limitations to consider when interpreting our results. While our sample size is comparable to those used in other studies [12,25], samples were selected to minimize confounding factors, and our power analysis suggests we should be able to capture significant changes to core bacteria composition, we could not account for significant variation associated with diet, life stages, and general environmental factors [9,70,71]. One potential environmental factor that may have impacted our results is the potential for co-infections. While our samples did not have multiple haemosporidian infections [37], infections by other pathogens, such as *Borrelia burgdorferi,* are possible [35]. We also encountered a significant challenge in obtaining high DNA extraction yields for sequencing. Although all samples included in this study were able to meet the criteria for sequencing, and cloacal swabs can be used as a safe proxy for gut microbiome studies [47,72], low yields could have limited our results and may explain some of the differences between our study and those of other studies [25–27]. Considering other studies were successfully able to collect high DNA yields with swabs [72], the relatively small size of the Junco likely contributed to this issue. Future studies on comparable or smaller birds could benefit from using fecal samples instead of the cloacal swabs as a proxy. The option to use 16S amplicon sequencing, as opposed to whole-genome sequencing, also impacts our results, as we can only assign functions based on taxonomic classification [73]. As a result, we are only able to describe differences in biodiversity rather than truly measure dysbiosis as it relates to loss of function, Lastly, our results are limited to male birds and may not be indicative of generalizations across all individuals [74].

## Conclusions

Parasite-microbiota interactions are an exciting avenue that may improve health outcomes for humans and animals. Our results show that infections by *Plasmodium* can correspond with significant changes to avian gut microbiota and several bacteria appear to be consistently affected. Considering these results remain relatively consistent between different host species, environmental conditions, and parasite lineages, it would be beneficial to delve deeper into the identity and functions provided by these bacteria. Our results suggest *Plasmodium* infections can correspond with characteristic metrics of dysbiosis, whereby the bacterial communities have lower diversity and lower abundance of beneficial taxa. However, it is unclear whether these changes adversely affect host health. Future studies can build on this work by determining if probiotic treatments of depleted bacterial species can affect infection susceptibility, whether prolonged changes to microbiota correspond with chronic detrimental health outcomes, and whether parasite virulence corresponds with the severity of gut microbiota community changes. As anthropogenically induced environmental change continues to alter disease dynamics, understanding the link between gut microbiota, parasitic infections, and host health can represent an essential avenue for supporting wildlife and human health.

## Acknowledgments

We want to acknowledge the contributions of all the undergraduate and graduate students in the Yeh lab from 2021–2023. In particular, Sara Freimuth and Austin Aguirre for assisting with the methodology. We would also like to thank Morgan Tingley, Tom Smith, and Justė Aželytė for their feedback, support, and comments on the analysis and manuscript preparation. We thank the UC NRS Santa Monica Mountains Reserve, UC campuses, and the National Park Service for land access.

## Author contributions

**Conceptualization:** Wilmer Stanley Amaya-Mejia, Ravinder NM Sehgal, Pamela J Yeh.

**Data curation:** Wilmer Stanley Amaya-Mejia.

**Formal analysis:** Wilmer Stanley Amaya-Mejia.

**Funding acquisition:** Wilmer Stanley Amaya-Mejia, Pamela J Yeh.

**Investigation:** Wilmer Stanley Amaya-Mejia.

**Methodology:** Wilmer Stanley Amaya-Mejia, Ravinder NM Sehgal.

**Resources:** Pamela J Yeh.

**Supervision:** Ravinder NM Sehgal, Pamela J Yeh.

**Visualization:** Wilmer Stanley Amaya-Mejia.

**Writing – original draft:** Wilmer Stanley Amaya-Mejia, Ravinder NM Sehgal.

**Writing – review & editing:** Wilmer Stanley Amaya-Mejia.

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
