## [Decision Letter · Decision Letter 0]

22 Oct 2025

*Junco hyemalis*

Dear Dr. Amaya-Mejia,

Thank you for submitting your manuscript to PLOS ONE. After careful consideration, we feel that it has merit but does not fully meet PLOS ONE’s publication criteria as it currently stands. Therefore, we invite you to submit a revised version of the manuscript that addresses the points raised during the review process.

We look forward to receiving your revised manuscript.

Kind regards,

Angela Monica Ionica, Ph.D.

Academic Editor

PLOS ONE

Journal Requirements:

3. Please expand the acronym “UCLA” (as indicated in your financial disclosure) so that it states the name of your funders in full.

4. We noted in your submission details that a portion of your manuscript may have been presented or published elsewhere. [Yes, a version of this manuscript is a part of WAM's doctoral dissertation and will need to be published on ProQuest as part of the graduation requirements.] Please clarify whether this publication was peer-reviewed and formally published. If this work was previously peer-reviewed and published, in the cover letter please provide the reason that this work does not constitute dual publication and should be included in the current manuscript.

6. Please note that PLOS One has specific guidelines on code sharing for submissions in which author-generated code underpins the findings in the manuscript. In these cases, we expect all author-generated code to be made available without restrictions upon publication of the work. Please review our guidelines at https://journals.plos.org/plosone/s/materials-and-software-sharing#loc-sharing-code and ensure that your code is shared in a way that follows best practice and facilitates reproducibility and reuse.

7. To comply with PLOS ONE submissions requirements, in your Methods section, please provide additional information regarding the experiments involving animals and ensure you have included details on methods of anesthesia and/or analgesia.

Reviewers' comments:

Reviewer's Responses to Questions

**Comments to the Author**

1. Is the manuscript technically sound, and do the data support the conclusions?

Reviewer #1: Yes

Reviewer #2: Yes

2. Has the statistical analysis been performed appropriately and rigorously?

Reviewer #1: Yes

Reviewer #2: Yes

3. Have the authors made all data underlying the findings in their manuscript fully available?

Reviewer #1: Yes

Reviewer #2: Yes

4. Is the manuscript presented in an intelligible fashion and written in standard English?

Reviewer #1: Yes

Reviewer #2: Yes

Reviewer #1: Summary:

This study by Amaya-Mejia and colleagues describes the alteration of gut microbiota in Juncos birds affected by Plasmodium. This paper is describing changes in microbiome associated to the infection, to discuss two main hypothesis about the link between dysbiosis and parasitic infection.

This study is well written, well conducted and very clear. It was a pleasure to read it, and I have no major comments. My only minor suggestions concern the text, and I expect no supplementary experiment.

Although clear, the Results section often lacks an interpretation or a comment after the key results. I made some suggestions hereafter. The introduction presents the challenge of the definition of dysbiosis, but the discussion lacks a follow-up to this introduction.

Please find hereafter my comments.

Abstract:

Line 19 : please rephrase the beginning of the sentence “For beta diversity, while infected birds had different bacteria than non-infected birds” as it is unclear

Line 28: the statement “However, in addition to the observed reduced microbial diversity, dysbiosis requires a significant increase in pathogenic bacteria and reduction of beneficial bacteria.” is unclear

Please use the same appellation for the birds between abstract and keywords, even if it refers to the same bird

Introduction:

Line 41 and later in the text: please either use the new names of Firmicutes (Bacillota) and Actinobacteria (Actinomycetes) or mention them

Line 44-47: could you discuss here the hypothesis of “the parasitic infection is permitted by dysbiosis”? You reject this hypothesis in the discussion, but it seems to me that it is missing from the introduction.

Line 48-50: can you add the notion of functions that are lost with dysbiosis?

Can you add a word about the consequences for Juncos of the parasite infection? Are they strictly carrier/vectors, do they suffer from the infection?

Last paragraph: not sure to follow how the different results would support one or the other hypothesis. This is clearer in the discussion part, notably with the “The hypothesis that gut microbiomes are altered due to Plasmodium infections is further supported by our co-occurrence networks that show significant dissimilarity between infected and non-infected birds, more than is expected by random chance.”, so please rephrase the introduction part to present more clearly how the results could support one or the other hypothesis.

Methods:

Could you provide the sequence of the primers used for the nested PCRs

Results:

The results section often lacks an interpretation or a comment after the key results. I made some suggestions hereafter.

Could you comment the following results “Based on microscopy, parasitemia for these eight infected birds ranged from 0.1 – 0.7% infected RBC.” indicating if the values correspond to what is expected?

Concerning body condition of the birds, could you explicit what is the body condition in terms of health for the bird? It would be clearer to me if you add a comment after the “Body condition, based on residuals of tarsus length and body mass, increased significantly with parasitemia” statement, to explicit the impact on the parasitemia on the bird health. If it’s relevant, you can also add a Figure showing the correlation.

Discussion:

The Discussion lacks a part about the definition of dysbiosis, and the limitation of using 16S sequencing compared to WGS. In the introduction, you discuss the definition of dysbiosis but never address it in the discussion. Moreover, the dysbiosis is a loss of microbial functions more than a decrease of diversity. The discussion is lacking this aspect, and this should be addressed in the limitations section as it is the main limitation of the study.

Line 311: “While Firmicutes have also been reported to be depleted in infected birds, this was not observed in our study.” The Figure 2 seems to indicate a decrease in Firmicutes. There is no comment about this result in the Results section, so could you add something about the non-significant difference that you propose?

In the Limitations section, please add a sentence about the fact that only males were used.

Figures:

Please either use the new names of Firmicutes (Bacillota) and Actinobacteria (Actinomycetes) or mention them

Typos:

Line 37: please remove the colon in the following sentence “In birds and other animals, these functions can include: nutritional uptake, detoxification, immune functions, and potentially serve to outcompete pathogenic microbes” to fit the syntax of the sentence

Line 55: please remove the can “but these can may not capture subtle variations and can be subject to sampling bias”

Line 57: please correct “it may be prudent to account for the relationships observed between bacteria, through co-occurrence networks”

Line 145: please correct “DNA extractions were also used”

Reviewer #2: Review report for manuscript entitled: Microbial community composition variation in response to malaria infections in Junco hyemalis

This manuscript explores the relationship between Plasmodium relictum GRW04 infection and gut microbiota composition in Junco hyemalis. The topic is bridging ecology, parasitology, and microbiome research. The study presents an interesting preliminary dataset and contributes to a growing understanding of parasite–microbiome interactions in wild avian systems.

The comments below are intended to support the authors in improving clarity, methodological robustness, and interpretative depth.

Major comments

- Sample size justification

The study includes only 16 individuals (8 infected, 8 uninfected). Authors mentioned that this sample size is similar to other used in animal studies. However, this requires at least some sort of statistical justification.

Authors acknowledge this limitation in the discussion, these results should be explicitly presented as exploratory.

Suggestion: Reframe all inferential language (e.g., “infection altered the microbiota”) to correlative phrasing (“infection status was associated with variation in…”).

- Use of microbiome instead of microbiota. In this study authors performed amplicon sequencing using 16S hypervariable regions V3-V4. Microbiota is the adequate word.

- Ambiguous causal interpretation between parasitemia and microbiome variation

Since all infections appear chronic and low intensity, microbiome variation might predate infection or reflect host condition rather than infection per se.

Suggestion: Rephrase interpretations to reflect association rather than causation. If possible, include parasitemia as a continuous covariate in multivariate analyses (rather than binary infection status) to test for dose-dependent effects.

- Framing and reporting inconsistencies

Reporting of statistical results is also inconsistent (some analyses lack test names or exact p-values).

Suggestion:

• Temper causal phrasing throughout (e.g., replace “infection caused dysbiosis” with “infection status was associated with variation of microbiota composition”).

• Provide full statistical reporting (test name, statistic, exact p-value) for all analyses.

Minor Comments

1. Abstract:

o Lines 22–23: Consider adding: rare bacterial taxa.

o Lines 24–26: The discussion of hypotheses may be better placed in the Discussion section rather than in the Abstract.

2. Introduction

o Baseline microbiome composition of Junco hyemalis. The introduction would benefit from a short paragraph summarizing known Junco or closely related passerine gut microbiome profiles.

o Ensure all bacterial phyla names are italicized throughout the manuscript.

o Line 52-54: extensive methodological details for introduction section. Consider removing.

o Line 55: Rephrase for clarity “but these can may not capture subtle variations…”

o

3. Methods

o Line 132–133: The phrase “Blood smears were periodically stained (within 30 days)” suggests repeated staining. Please clarify whether staining occurred once per sample and specify the time between fixation and staining.

o Line 145–148: Replace the second repeated phrase “nested PCR approach” with “This technique.”

o PCR protocol description: it is more useful, for reproducibility purpose, to report concentrations of the elements included rather than volumes. Please also include the exact number of cycles both PCR rounds.

o There is no mention of verifying DNA concentration or purity for blood samples prior to nested PCR amplification.

o Line 150: Replace “H20” with “H₂O” (letter O, not zero).

o Use the proper microliter symbol “µL” in all instances.

o Only define abbreviations the first time they appear on the text.

o Consider adding a brief paragraph outlining the Sanger sequencing protocol (e.g., cleanup, sequencing platform, read quality verification).

o Line 156–157: Reword for clarity

o Line 168: Provide the evaporation temperature and duration during DNA concentration.

o The manuscript authors refer to “low parasitemia” but does not specify parasitemia scale defining low vs high infection.

o Line 174: already mentioned in data availability section no need to mention it here.

4. Results

o Line 205: how many individuals were screened in total ?

o Table 1: Move the explanatory text (“Parasitemia is based on...”) to the table footnote.

o Table 1: Replace “0/1” infection status code with “No detectable / Detectable” for clarity.

o Table1: Does banding date corresponds with sample collection date? If so could be useful to have a column of DNA extraction date.

o Confirm whether banding date corresponds to the date of sample collection and, if relevant, add a column indicating DNA extraction date.

o Figure 2b: Replace “bacteria” with “phyla” in the legend.

o Consider visualizing the ANCOM-BC results in a volcano plot. The following paper might help: Maust et al. (DOI: https://doi.org/10.1016/j.heliyon.2023.e22145)

o Figure 3: The legend should mention the type of ordination used (e.g., PCoA) and the percentage of variance explained on each axis. Please revise the legend.

o Line 225: ASV abbreviation redefined.

o Line 240: Plasmidium relictum is not italicized.

5. Discussion

o State whether non-Plasmodium-infected birds were screened for other pathogens that could confound microbiota results.

**Do you want your identity to be public for this peer review?** For information about this choice, including consent withdrawal, please see our Privacy Policy

Reviewer #1: No

Reviewer #2: No

---

## [Author Response · Author response to Decision Letter 1]

3 Dec 2025

Please find below a response to review and editor comments. The response is also included as a separate Word document attached to this submission with color-coordinated responses for ease of review.

On behalf of all of the authors, thank you,

Wilmer

Comments to Editors

The manuscript has been updated to meet the PLOS ONE style requirements.

The Financial Disclosure section has now been updated. The grants received do not include specific grant numbers.

3. Please expand the acronym “UCLA” (as indicated in your financial disclosure) so that it states the name of your funders in full.

The acronym has now been updated.

4. We noted in your submission details that a portion of your manuscript may have been presented or published elsewhere. [Yes, a version of this manuscript is a part of WAM's doctoral dissertation and will need to be published on ProQuest as part of the graduation requirements.] Please clarify whether this publication was peer-reviewed and formally published. If this work was previously peer-reviewed and published, in the cover letter please provide the reason that this work does not constitute dual publication and should be included in the current manuscript.

The content was not peer-reviewed and therefore does not constitute a dual publication.

5. Review Comments to the Author

Reviewer #1: Summary:

This study by Amaya-Mejia and colleagues describes the alteration of gut microbiota in Juncos birds affected by Plasmodium. This paper is describing changes in microbiome associated to the infection, to discuss two main hypothesis about the link between dysbiosis and parasitic infection.

This study is well written, well conducted and very clear. It was a pleasure to read it, and I have no major comments. My only minor suggestions concern the text, and I expect no supplementary experiment.

Although clear, the Results section often lacks an interpretation or a comment after the key results. I made some suggestions hereafter. The introduction presents the challenge of the definition of dysbiosis, but the discussion lacks a follow-up to this introduction.

Please find hereafter my comments.

Thank you for your feedback and recommendations on how we can improve this manuscript.

Abstract:

Line 19 : please rephrase the beginning of the sentence “For beta diversity, while infected birds had different bacteria than non-infected birds” as it is unclear

This section is now revised to improve clarity. Line 21.

Line 28: the statement “However, in addition to the observed reduced microbial diversity, dysbiosis requires a significant increase in pathogenic bacteria and reduction of beneficial bacteria.” is unclear

This section was removed to improve clarity.

Please use the same appellation for the birds between abstract and keywords, even if it refers to the same bird

The keywords and text has updated to use the name “Oregon Junco”. Line 31.

Introduction:

Line 41 and later in the text: please either use the new names of Firmicutes (Bacillota) and Actinobacteria (Actinomycetes) or mention them

Thank you for this correction, we have decided to use the updated names Bacillota and Actinomycetes through the manuscript. Line 40, 41, Figure 2, Figure 5

Line 44-47: could you discuss here the hypothesis of “the parasitic infection is permitted by dysbiosis”? You reject this hypothesis in the discussion, but it seems to me that it is missing from the introduction.

We have now added a section outlining the potential role of dysbiosis related to host health outcomes and specifically an increased susceptibility to parasitic infections. Line 44.

Line 48-50: can you add the notion of functions that are lost with dysbiosis?

Can you add a word about the consequences for Juncos of the parasite infection? Are they strictly carrier/vectors, do they suffer from the infection?

We now have expanded our introduction to address the significance of dysbiosis and how it may affect functions. Line 50.

Last paragraph: not sure to follow how the different results would support one or the other hypothesis. This is clearer in the discussion part, notably with the “The hypothesis that gut microbiomes are altered due to Plasmodium infections is further supported by our co-occurrence networks that show significant dissimilarity between infected and non-infected birds, more than is expected by random chance.”, so please rephrase the introduction part to present more clearly how the results could support one or the other hypothesis.

We have revised our introduction to better reflect the hypotheses, specifically as it relates to our hypothesis outlining the possibility of malaria-associated dysbiosis. Line 78-81.

Methods:

Could you provide the sequence of the primers used for the nested PCRs

We have now added the primer sequences. Line 163-165

Results:

The results section often lacks an interpretation or a comment after the key results. I made some suggestions hereafter.

Based on the comments provided, we have revised our manuscript to provide interpretation throughout.

Could you comment the following results “Based on microscopy, parasitemia for these eight infected birds ranged from 0.1 – 0.7% infected RBC.” indicating if the values correspond to what is expected?

We have now added commentary explaining that we primarily observe ~1% parasitemia but 50% has been observed in certain systems. Line 243.

Concerning body condition of the birds, could you explicit what is the body condition in terms of health for the bird? It would be clearer to me if you add a comment after the “Body condition, based on residuals of tarsus length and body mass, increased significantly with parasitemia” statement, to explicit the impact on the parasitemia on the bird health. If it’s relevant, you can also add a Figure showing the correlation.

We have now added additional context that suggested the observed difference in body condition is potentially due to sample size but explain how a higher body condition may also still be detrimental depending on how body mass is distributed. Line 245.

Discussion:

The Discussion lacks a part about the definition of dysbiosis, and the limitation of using 16S sequencing compared to WGS. In the introduction, you discuss the definition of dysbiosis but never address it in the discussion. Moreover, the dysbiosis is a loss of microbial functions more than a decrease of diversity. The discussion is lacking this aspect, and this should be addressed in the limitations section as it is the main limitation of the study.

Thank you for your feedback, we have updated our Discussion to emphasize the significance of dysbiosis and the impact on functions. Line 414. We have also included an acknowledgement of how 16S sequencing may be insufficient to capture impacts on function. Line 436.

Line 311: “While Firmicutes have also been reported to be depleted in infected birds, this was not observed in our study.” The Figure 2 seems to indicate a decrease in Firmicutes. There is no comment about this result in the Results section, so could you add something about the non-significant difference that you propose?

We have now added to our Results section explaining that observed differences in taxa were not statistically significant. In addition, we mention how broad changes may be difficult to observe, specifically in birds. Line 359.

In the Limitations section, please add a sentence about the fact that only males were used.

Done. Line 439.

Figures:

Please either use the new names of Firmicutes (Bacillota) and Actinobacteria (Actinomycetes) or mention them

Names for these taxa have now been updated.

Typos:

Line 37: please remove the colon in the following sentence “In birds and other animals, these functions can include: nutritional uptake, detoxification, immune functions, and potentially serve to outcompete pathogenic microbes” to fit the syntax of the sentence

Done. Line 36.

Line 55: please remove the can “but these can may not capture subtle variations and can be subject to sampling bias”

This line was removed following revisions.

Line 57: please correct “it may be prudent to account for the relationships observed between bacteria, through co-occurrence networks”

This line was removed following revisions.

Line 145: please correct “DNA extractions were also used”

This line was updated following revisions. Line 159.

Reviewer #2: Review report for manuscript entitled: Microbial community composition variation in response to malaria infections in Junco hyemalis

This manuscript explores the relationship between Plasmodium relictum GRW04 infection and gut microbiota composition in Junco hyemalis. The topic is bridging ecology, parasitology, and microbiome research. The study presents an interesting preliminary dataset and contributes to a growing understanding of parasite–microbiome interactions in wild avian systems.

The comments below are intended to support the authors in improving clarity, methodological robustness, and interpretative depth.

Thank you for your feedback and highlighting areas for improvement. We have provided our response below to improve the manuscript.

Major comments

- Sample size justification

The study includes only 16 individuals (8 infected, 8 uninfected). Authors mentioned that this sample size is similar to other used in animal studies. However, this requires at least some sort of statistical justification.

We have now added additional support based on power analysis that would be sufficient to determine a major shift in the relative abundance of Bacillota (updated from Firmicutes) based on change measured in other studies. Line 185.

Authors acknowledge this limitation in the discussion, these results should be explicitly presented as exploratory.

We have added a sentence specifically stating that these results are exploratory. Line 238.

Suggestion: Reframe all inferential language (e.g., “infection altered the microbiota”) to correlative phrasing (“infection status was associated with variation in…”).

We agree with this recommendation and have revised our manuscript to highlight the correlation rather than causality.

- Use of microbiome instead of microbiota. In this study authors performed amplicon sequencing using 16S hypervariable regions V3-V4. Microbiota is the adequate word.

We have now corrected this language. Ex: Line 18, Line 107, Line 109, Line 239, Line 348.

- Ambiguous causal interpretation between parasitemia and microbiome variation

Since all infections appear chronic and low intensity, microbiome variation might predate infection or reflect host condition rather than infection per se.

Suggestion: Rephrase interpretations to reflect association rather than causation. If possible, include parasitemia as a continuous covariate in multivariate analyses (rather than binary infection status) to test for dose-dependent effects.

We agree with this comment and have revised the language to more accurately reflect that the relationships observed are only associative. We have also performed supplemental analysis with parasitemia, rather than only binary status, to assess any dose effect. We did not find any relationship based on our samples. Line 216, Line 276.

- Framing and reporting inconsistencies

Reporting of statistical results is also inconsistent (some analyses lack test names or exact p-values).

We have now added the test names and added the exact adjust p-values throughout. Line 246, 261, 268, 269, 271, 274, 275, 277, 297, 302, 315, 317, 318, 319, 320.

Suggestion:

• Temper causal phrasing throughout (e.g., replace “infection caused dysbiosis” with “infection status was associated with variation of microbiota composition”).

The language has now been adjusted to emphasize associations not causality.

• Provide full statistical reporting (test name, statistic, exact p-value) for all analyses.

Statistics have now been added.

Minor Comments

1. Abstract:

Lines 22–23: Consider adding: rare bacterial taxa.

Done. Line 20.

Lines 24–26: The discussion of hypotheses may be better placed in the Discussion section rather than in the Abstract.

This section has now been removed from the abstract.

2. Introduction

Baseline microbiome composition of Junco hyemalis. The introduction would benefit from a short paragraph summarizing known Junco or closely related passerine gut microbiome profiles.

We have now added a paragraph outlining research on juncos/sparrows. Line 94 -106.

Ensure all bacterial phyla names are italicized throughout the manuscript.

Done.

Line 52-54: extensive methodological details for introduction section. Consider removing.

This section has now been removed.

Line 55: Rephrase for clarity “but these can may not capture subtle variations…”

This line was removed following revisions.

3. Methods

Line 132–133: The phrase “Blood smears were periodically stained (within 30 days)” suggests repeated staining. Please clarify whether staining occurred once per sample and specify the time between fixation and staining.

This section has been corrected to reflect that the staining was only performed once (within 30 days) and that fixing with methanol was performed within a few hours of collection. Line 145.

Line 145–148: Replace the second repeated phrase “nested PCR approach” with “This technique.”

Done.

PCR protocol description: it is more useful, for reproducibility purpose, to report concentrations of the elements included rather than volumes. Please also include the exact number of cycles both PCR rounds.

We have now added this information as requested. Line 164 – 169.

There is no mention of verifying DNA concentration or purity for blood samples prior to nested PCR amplification.

We apologize for this oversight. DNA concentration and purity was assessed using a Nanodrop. This information has now been added. Line 159.

Line 150: Replace “H20” with “H₂O” (letter O, not zero).

This error has now been corrected. Line 168.

Use the proper microliter symbol “µL” in all instances.

This error has now been corrected. Line 167-170

Only define abbreviations the first time they appear on the text.

Done.

Consider adding a brief paragraph outlining the Sanger sequencing protocol (e.g., cleanup, sequencing platform, read quality verification).

This information has now been added. Samples underwent clean up with ExoSAP-IT, sequenced by GENEWIZE using the Applied Biosystems platform, and reads were QCed to ensure a minimum of 80% high-quality. Line 170 – 175.

Line 156–157: Reword for clarity

Revised. Line 176.

Line 168: Provide the evaporation temperature and duration during DNA concentration.

We have now added the temperature (37C) and time, specifically noting that we aimed for 10-minute intervals to reach a volume of 50µL. Line 191.

The manuscript authors refer to “low parasitemia” but does not specify parasitemia scale defining low vs high infection.

We apologize for this oversight. We have now clarified that “low” is relative to reported levels of parasitemia (50%) but that this is relatively consistent based on previous observations. Line 243.

Line 174: already mentioned in data availability section no need to mention it here.

Removed.

4. Results

Line 205: how many individuals were screened in total ?

We originally screened 151 adult juncos, this has now been added. Line 141.

Table 1: Move the explanatory text (“Parasitemia is based on...”) to the table footnote.

Don

---

## [Decision Letter · Decision Letter 1]

23 Dec 2025

Microbial community composition variation in relation to malaria infections in *Junco hyemalis*

PONE-D-25-49187R1

Dear Dr. Amaya-Mejia,

We’re pleased to inform you that your manuscript has been judged scientifically suitable for publication and will be formally accepted for publication once it meets all outstanding technical requirements.

Kind regards,

Angela Monica Ionica, Ph.D.

Academic Editor

PLOS One

Additional Editor Comments (optional):

Reviewers' comments:

Reviewer's Responses to Questions

**Comments to the Author**

Reviewer #1: All comments have been addressed

Reviewer #2: All comments have been addressed

2. Is the manuscript technically sound, and do the data support the conclusions?

Reviewer #1: Yes

Reviewer #2: Yes

3. Has the statistical analysis been performed appropriately and rigorously?

Reviewer #1: Yes

Reviewer #2: Yes

4. Have the authors made all data underlying the findings in their manuscript fully available?

Reviewer #1: Yes

Reviewer #2: Yes

5. Is the manuscript presented in an intelligible fashion and written in standard English?

Reviewer #1: Yes

Reviewer #2: Yes

Reviewer #1: (No Response)

Reviewer #2: Authors addressed all suggested comments. The manuscript is clear and concise with sufficient methodological detail to be reproduced by the scientific community. I have no further recommendations.

**Do you want your identity to be public for this peer review?** For information about this choice, including consent withdrawal, please see our Privacy Policy

Reviewer #1: No

Reviewer #2: No

---

## [Editor Report · Acceptance letter]

PONE-D-25-49187R1

PLOS One

Dear Dr. Amaya-Mejia,

I'm pleased to inform you that your manuscript has been deemed suitable for publication in PLOS One. Congratulations! Your manuscript is now being handed over to our production team.

Kind regards,

on behalf of

Dr. Angela Monica Ionica

Academic Editor

PLOS On